# Existence of a Neutral-Impact Maxillo-Mandibular Displacement on Upper Airways Morphology

**DOI:** 10.3390/jpm11030177

**Published:** 2021-03-04

**Authors:** Giovanni Badiali, Ottavia Lunari, Mirko Bevini, Barbara Bortolani, Laura Cercenelli, Manuela Lorenzetti, Emanuela Marcelli, Alberto Bianchi, Claudio Marchetti

**Affiliations:** 1Department of Maxillo-Facial Surgery, IRCCS Azienda Ospedaliero-Universitaria di Bologna, 40138 Bologna, Italy; giovanni.badiali@unibo.it (G.B.); mirko.bevini@studio.unibo.it (M.B.); claudio.marchetti@unibo.it (C.M.); 2Department of Biomedical and Neuromotor Sciences, University of Bologna, 40138 Bologna, Italy; 3eDIMES Lab-Laboratory of Bioengineering, Department of Experimental Diagnostic and Specialty Medicine, University of Bologna, 40138 Bologna, Italy; barbara.bortolani@unibo.it (B.B.); laura.cercenelli@unibo.it (L.C.); manuela-luca@libero.it (M.L.); emanuela.marcelli@unibo.it (E.M.); 4Department of Surgery and Surgical Specialties, Azienda Ospedaliero-Universitaria Rodolico-S. Marco, University of Catania, 95124 Catania, Italy; alberto.bianchi@unict.it

**Keywords:** upper airways, orthognathic surgery, maxillofacial surgery, 3D analysis method, OSAS, bioengineering

## Abstract

Current scientific evidence on how orthognathic surgery affects the airways morphology remains contradictory. The aim of this study is to investigate the existence and extension of a neutral-impact interval of bony segments displacement on the upper airways morphology. Its upper boundary would behave as a skeletal displacement threshold differentiating minor and major jaw repositioning, with impact on the planning of the individual case. Pre- and post-operative cone beam computed tomographies (CBCTs) of 45 patients who underwent maxillo-mandibular advancement or maxillary advancement/mandibular setback were analysed by means of a semi-automated three-dimensional (3D) method; 3D models of skull and airways were produced, the latter divided into the three pharyngeal subregions. The correlation between skeletal displacement, stacked surface area and volume was investigated. The displacement threshold was identified by setting three ∆Area percentage variations. No significant difference in area and volume emerged from the comparison of the two surgical procedures with bone repositioning below the threshold (approximated to +5 mm). A threshold ranging from +4.8 to +7 mm was identified, varying in relation to the three ∆Area percentages considered. The ∆Area increased linearly above the threshold, while showing no consistency in the interval ranging from −5 mm to +5 mm.

## 1. Introduction

The existence of a strong correlation between maxillary bones repositioning, upper airways and craniofacial morphology is currently widely assumed among researchers and clinicians, owing to early studies proving that orthognathic surgery could effectively alter the configuration of the pharyngeal space [1]. Initially, the evaluation of the upper airways (UA) was mainly conducted bidimensionally on lateral cephalometric radiographs, but lacked sufficient accuracy in representing three-dimensional (3D) anatomical structures. Due to the rise in popularity of cone beam computed tomography (CBCT), the focus of the vast majority of research has increasingly shifted toward a three-dimensional assessment of the pharynx [2,3].

However, despite the existence of widely recognised protocols for the reproducibility of imaging acquisition for standardization purposes [4,5], CBCTs appear liable to invalidation [6]. Additionally, a consensus has yet to be reached in regard to the definition of a consistent method for the UA morphological evaluation, although different attempts were pursued [7,8]. To our knowledge, the current literature lacks studies investigating both morphological and volumetric variations of the airways contextually comparing maxillo-mandibular advancement and maxillary advancement/mandibular setback in correlation to the quota of jaw displacement performed, whilst striving to overcome the controversies embedded in imaging acquisition and anatomical landmarks position reproducibility.

The aim of this research is to investigate whether maxillo-mandibular advancement and maxillary advancement/mandibular setback elicit a significantly different impact on the UA morphology. Expanding on preliminary results, we set out to determine the existence and extension of a neutral-impact interval of maxillary and mandibular displacement, within which the UA morphology exhibits only minor and unpredictable variations. The upper bound of this range could be considered as a skeletal displacement threshold, which would allow us to discriminate between ‘minor’ and ‘major’ jaw repositioning in regard to UA area and volume variations, in both surgical procedures. The lower bound of the said range would represent a displacement below which volume and cross-sectional area of the UA would show a decreasing trend.

In other words, the purpose of the authors is to challenge the established null hypothesis that maxillo-mandibular advancement leads to UA area and volume increase, while maxillary advancement with mandibular setback to UA area and volume decrease. We hereby introduce an automated 3D analysis that, by correlating jaw repositioning and shape parameters of the UA, may provide information useful for predictive modeling of UA morphological variation, to be applied in the virtual surgical planning of the individual case. In this paper we present the 3D analysis method together with the results obtained on a cohort of 45 patients.

## 2. Materials and Methods

### 2.1. Ethics Statement

The ethical approval for the present protocol has been granted by the Sant’Orsola-Malpighi University Hospital ethics committee (approval no. 313/2019/Oss/AOUBO) on 22 May 2019. The study conformed to the principles of the Helsinki Declaration. Participating patients were informed about the study and provided written informed consent upon enrollment. Data were collected pseudo-anonymously and patients were assigned a progressive code. Data were then archived anonymously so that patients could be identified exclusively by their assigned code.

### 2.2. Patient Population

A retrospective evaluation was conducted of 45 patients—mean age 25.5 years (range 18–51 years)—undergoing orthognathic surgery between January 2010 and April 2019 at the Oral and Maxillo-Facial Surgery Unit of the S. Orsola-Malpighi University Hospital (Bologna, Italy). All procedures were performed by the same surgeon. Pre-operative (T1, one month prior to surgery) and post-operative (T2, 7.5 ± 1 months after surgery) CBCT scans were analysed.

All of the selected 45 patients met the following inclusion criteria: (a) presence of class II or class III dento-skeletal dysgnathia, (b) complete pre-operative and post-operative diagnostic 3D study, (c) pre- and post-operative CBCTs presenting the highest imaging quality, allowing us to precisely and easily identify the anatomical landmarks of interest. Patients presenting cleft lip and palate, craniofacial syndromes and pre-existent trauma were excluded from the study. Polysomnographic data were also acquired for obstructive sleep apnea syndrome (OSAS) patients. The included patients were divided into two subgroups according to the performed surgical procedure: Group 1, maxillo-mandibular advancement—MMA, (*n* = 29, 11 males, 18 females, mean age 26.5 years); Group 2, maxillary advancement associated to mandibular setback—MAMS, (*n* = 16, 6 males, 10 females, mean age 23.8 years).

### 2.3. Data Acquisition

Patients were enrolled once pre-operative orthodontic treatment was completed, or an adequate and stable final occlusion was achieved. All patients performed CBCT scans (NewTom VGI EVO, Cefla Group, Imola, Italy), (24 × 19 cm FOV, 0.3 mm voxel) in a clinically determined natural head position (NHP) and using a wax bite obtained in clinically set condylar centric relation (with the aid of Dawson’s maneuver), according to Guijarro-Martinez’s and Swennen’s [9] guidelines for airways evaluation.

## 3. Study Design

### 3.1. Image Segmentation

To obtain 3D reconstructions of the anatomical regions of interest (skull, maxillary bones and upper airways), CBCT datasets were exported in DICOM format and digitally processed. Facial hard tissue and the upper airways were semi-automatically segmented using 3D Slicer software (open source software, www.slicer.org, accessed on 1 March 2021), the mean segmentation thresholds being +646.19 (± 24.57) to +1000 Hounsfield for hard tissue, while being −1000 to 299.39 (± 17.2) Hounsfield for upper airways. The UA were then cropped via the Crop volumes tool. The superior border was defined by the perpendicular intersection of two planes: tangent to the occipital basion and tangent to the most prominent point of the sphenoid sinus (SpS), the latter also marked the anterior limit. Inferiorly, the pharynx was cut at the inferior edge of C5. Digital models were exported to STL files (Figure 1a,b).

### 3.2. Pre- to Post-Operative Model Registration

The STL files were loaded onto the open-source software CloudCompare (MacOs version 2.6.1, Cloud Compare project, www.cloudcompare.org, accessed on 1 March 2021). The pre- to post-operative bone displacement and the morphological readjustment of the airways of each patient were determined according to the following procedure:Alignment of 3D cranial base models through the Align (point pairs picking) tool (Tool > Registration > Align). Specifically, two pairs of landmarks unaltered by surgery were manually defined: most medial point of the frontozygomatic sutures bilaterally; most concave point of the inner posterior border of each zygomatic arch. The pre-operative mesh was defined as the Reference entity (fixed), the post-operative as Aligned (moving). The alignment of the pre- to post-operative mesh was then refined via iterative closest point (ICP) alignment; reference and aligned entities were set as in the previous step. Finally, a colorimetric surface map was generated to visually inspect the obtained superimposition for misalignments due to segmentation artifacts.Digital model sectioning to obtain a detached upper maxilla and mandible.Point-by-point alignment of the bone sections via four point pairs picking for each bony segment considered: most mesio-vestibular cusp of the first molar bilaterally and canine cusp bilaterally. The alignment was refined via ICP following the aforementioned workflow. The transformation matrices referring to each bony fragment were analysed to express the corresponding rigid body transformation (i.e., resulting bone displacement) in terms of Euler angles convention, which is composed of three angles (pitch, roll, yaw) and three translations (x, y, z, or lateral, anteroposterior and craniocaudal). The signed discrepancies were tabulated applying conventional signs.Point-by-point manual alignment of the upper airways models via three point pairs (pharyngeal recess bilaterally, interarithenoid notch) followed by ICP refinement and colorimetric surface map inspection. (Figure 2a–c)

### 3.3. Three-Dimensional (3D) Analysis

Starting from the 3D models of UA and bony segments that were obtained, the following 3D analysis was carried out. All calculations were performed using MatLab software (version R2019a, Mathworks, Natick, MA, USA).
Partitioning of the UA models into 25 bounding boxes, defined as the smallest volume able to contain a set of points, with sides parallel to the three axes of the reference system. (Figure 3)Subdivision of the pharynx into three sectors according to its anatomical regions: naso-, oro-, hypopharynx. Each region was assigned a corresponding number of Bounding Boxes (5, 9, 11; respectively).Computation of the projected surface area (mm^2^) of the UA section delimited by each bounding box.Computation of the total volume (mm^3^) of the pharyngeal airway space.

The fixed UA segmentation landmarks and number of bounding boxes were selected in order to perform an automated normalisation of the vertical measure of the UA to compensate for post-operative height modifications.

The variation of the surface area of each section was calculated as percentage ∆Area.

All measures were clustered according to the pharyngeal subregions and summarised in mean and standard deviation, for both groups (MMA; MAMS). Maxillary bones displacement was correlated to ∆Area for the three anatomical regions. The analysis of the variation of UA total volume was carried out as ∆ percentage volume. A comprehensive analysis of spatial displacement of bony segments (x, y, z translation; yaw, pitch, roll) was conducted. However, in this study only antero–posterior repositioning (AP) was considered, due to the complexity associated to the interpretation of multi-varied analysis.

### 3.4. Statistical Analysis

Based on the inspection of the plotted data of all 45 patients (Figure 4a,b), we observed a displacement threshold around +5 mm for each bony segment. Below this point, both upper airways ∆Area increases and ∆Area decreases were found, thus not correlating to maxillary bones displacement. Contrarily, when the +5 mm displacement was met, the ∆Area increased in a more linear fashion, in correlation to the jaw displacement, except for three cases. Hence, to confirm the existence of the +5 mm threshold, we initially ran a one-way variance analysis (ANOVA) and a post hoc Games–Howell test.

We compared the following subgroups to investigate whether the two surgical procedures (MAMS, MMA) would lead to significantly different ∆Area variation:
MAMS vs. MMA with Upper Maxilla AP < 5 mmMAMS vs. MMA with Mandible AP < 5 mmMAMS vs. MMA with Upper Maxilla AP ≥ 5 mmMAMS vs. MMA with Mandible AP ≥ 5 mm

A non-parametric Mann–Whitney test was performed to compare the UA ∆ total Volume percentage among the same subgroups. A *p* value < 0.05 was considered statistically significant.

To test the consistency of the threshold value, we set the cut-off to +100%, +50% and +30% ∆Area increase. Receiver operating characteristic (ROC) curves were traced to statistically determine the presence of the maxillary segments’ displacement threshold and its pattern of variation in accordance with the ∆Area increase. The data were evaluated both collectively and separately, divided according to the surgical procedure. Jaw displacement of upper maxilla and mandible were independently correlated to each subregion of the pharynx.

To express the overall impact of the combined maxillary bones displacement on the ∆Area increase, multi-variate ROC curves were calculated using the value of logistic regression resulting from the combined displacement of the maxillary and mandibular segments of each single case (MMA; MAMS, independently). This analysis was carried out only for the +30% ∆Area increase, due to a positive-to-negative ratio of slices closer to 1.

Youden’s index (J) was used to assess the cut-off. The index represents the sum of sensitivity and specificity (minus one) and reflects the performance of a dichotomous test at given value of a variable (in this case the AP displacement). The maximum value of Youden’s index expresses the displacement threshold point. The displacement values (mm) corresponding to the threshold identified by Youden’s index were tabulated.

Data analysis was performed with IBM SPSS 25 (IBM, Armonk, NY, USA). All tests were two-tailed. For all tests the significance level was set to α = 0.05.

## 4. Results

Descriptive statistics are reported in Table 1 and Table 2. Median upper maxillary and mandibular bones AP displacement performed in MMA group were +3.8 and +4.6 mm, respectively. Median values in MAMS group are +3.9 and −2.5 mm, respectively.

### 4.1. Analysis of Variance (ANOVA) Test

One-way ANOVA test found a significant difference in ∆Area between the considered groups at the level of all three subregions, naso-, oro-, hypopharynx (*p* < 0.001; *p* < 0.001; *p* < 0.001, respectively).

### 4.2. Games–Howell Test

MAMS compared to MMA with upper maxilla AP movement <5 mm proved to be not significantly different along all three subregions (*p* = 0.973; 0.103; 1.000, respectively). MAMS to MMA with mandible AP <5 mm similarly shows no statistical difference (*p* = 0.998; 1.000; 0.993). By contrast, MAMS to MMA with upper maxilla AP ≥5 mm proved to be significantly different (*p* < 0.001), as well as MAMS to MMA with mandible AP ≥5 mm (*p* < 0.001).

### 4.3. Mann–Whitney Test

∆ percentage volume MAMS vs. MMA with upper maxilla AP <5 mm and MAMS vs. MMA with mandible AP <5 mm were found to be not significantly different (*p* = 1.000, both cases).

MAMS vs. MMA with upper maxilla AP ≥5 mm and MAMS vs. MMA with mandible AP ≥5 mm were significantly different (*p* < 0.005).

### 4.4. Receiver Operating Characteristic (ROC) Curves

The data of Receiver Operating Characteristic (ROC) Curves are reported in Table 3.

#### 4.4.1. Maxilla

A +100% ∆Area increase was obtained with a cut-off between +4.8 mm (hypopharynx) and +7 mm (oropharynx) of maxillary AP displacement. The +50% and +30% ∆Area increase cut-off comprised between +4.8 mm and +6.6 mm of maxillary AP advancement (Figure 5A–C).

#### 4.4.2. Mandible

A +100% ∆Area increase was obtained with a cut-off between +5.3 mm (hypopharynx) and +6.3 mm (oropharynx) of maxillary AP displacement. The +50% and +30% ∆Area increase cut-off comprised between +5.2 mm and +6 mm of mandible advancement (Figure 6A–C).

Both in +50% and +30% ∆Area increase we obtained a dual peak in Youden’s index value, suggesting the existence of a double threshold. The threshold point tabulated outside of brackets is the one both statistically and clinically more consistent with the cut-points found in the rest of the analysis. The other threshold point is found at 1.4 mm of advancement in both curves.

Due to the complex interpretation of the combined maxillary and mandibular displacement ROC curve, the analysis was used to select a series of cases in which the ∆Area increase respected the given parameters (>30%) in each defined subregion. The characteristics of displacement of each case were qualitatively evaluated in the search for clinically significant maxillary and mandibular displacement combinations.

## 5. Discussion

Although an increasing number of studies have evaluated how orthognathic surgery affects the upper airways, to our knowledge none of the existing methods proved to be fully reliable, as per heterogenous methodological shortcomings of varying criticality. Several proposed methods of pre- to post-operative skull and airways 3D models proved to be unreliable due to lack of control over respiratory phase, tongue posture, head and neck inclination, as well as poor intra- and inter-examiner reliability [10,11].

Moreover, anatomical regions either direct displacement or being affected by maxillary bones repositioning (e.g., anterior nasal spine (ANS), posterior nasal spine (PNS), epiglottis) were inaccurately selected as fixed landmarks to assess the UA variation [12,13,14,15,16,17]. In this regard, Muto et al. [18] reported that an increase of 10° in cranio-cervical inclination at the level of C2 augmented the pharyngeal airway space (PAS) of approximately 4 mm.

Subsequently, also cephalometric planes proved to be misleading when used as a reference for the assessment of the upper airways evaluation. Maxillary repositioning not only affects the volume of the pharynx, but its shape as well [19,20], due to the morphological readjustment induced by the surrounding anatomical structures. Therefore, if the Frankfurt plane were to be used as a reference plane, it would result in an alignment of morphologically incomparable airways.

We opted for a direct alignment of the models via point pairs picking and ICP refinement based on the assumption that, regardless of the existence of protocols for the standardization of CBCT acquisition, any minimal patient-related heterogeneity could invalidate the superimposition of post- to pre-operative 3D skull and UA models. As a result, the posterior wall of the pharynx was chosen for the alignment of the airways models as it undergoes negligible modification following maxillary repositioning.

The current body of literature is heterogeneous in terms of analytical methods as it is in terms of results obtained by the said analyses. The most widely accepted assumption is that a certain bony fragment displacement (mainly along the antero–posterior axis) elicits a comparable variation, be it contraction or dilation, of the airways space immediately behind it.

By contrast, upon plotting, our data showed a skeletal displacement threshold, initially approximated to 5 mm (Figure 4), below which the airway space did not appear to vary consistently with the skeletal displacement. The results of the ANOVA and Games–Howell tests confirm the validity of the said threshold value for both bony fragments analysed. To note, after producing ROC curves, the mean displacement threshold at 50% ∆area increase between all slices of the three pharynx subregions was 5.53 mm. The authors are aware of the discrepancy between this value and the 5 mm threshold; however, this discrepancy was accepted in order to run the preliminary statistics. Also, in surgical terms, a difference of 0.5 mm in displacement is hardly controllable.

Games–Howell test results point to the fact that there is no significant difference in UA ∆area between MMA and MAMS with maxillary bones AP movement lesser than 5 mm, suggesting that moderate mandibular setbacks with concomitant upper jaw advancement negligibly affect the UA morphology. Following a similar logic, it may also suggest that mild bimaxillary advancement (less than +5 mm) does not induce consistent expansions of the UA.

Notably, one case reported a ∆area increase of 50% at the level of the nasopharynx following MAMS (mandibular setback: −3.68 mm; maxillary advancement: +7.2 mm). It is reasonable to hypothesize that a higher degree of upper maxillary forward translation counteracts the reduction in surface area that mandibular setback may induce, as previously reported [21,22].

Similarly to the Games–Howell’s test, the Mann–Whitney test highlights a lack of statistical difference in ∆ total volume percentage when MMA is compared to MAMS with AP <5 mm, reiterating the results obtained from the evaluation of ∆area variation. Ultimately, these findings contrast with previous results, such as those reported by Christovam [23].

Recently, other studies have reported that mandibular setback, both alone and concomitant with upper maxillary advancement, may not induce a reduction of upper airways space [24]. However, due to methodological biases, a direct comparison with our study could not be carried out.

In regards to the cut-off values, +30% and +50% ∆Area increases require comparable skeletal displacements in order to be achieved, for both maxilla and mandible, along all three subregions of the pharynx. Values ≥100% of ∆Area increase were shown to require higher degrees of displacement of the maxilla at naso- and oropharynx level and of the mandible at the hypopharynx. These findings suggest that an expansion of the UA area greater than +50% requires the AP displacement threshold to follow a linear trend, thus greater maxillary bone movement needs to be performed to more strongly correlate to the aimed ∆Area increase.

Intuitively, the positive-to-negative ratio of UA sections analysed via ROC sensibly decreases when greater area expansions are considered. In other words, +30% ∆Area increase value shows the highest number of positive slices and +100% the lowest. Given the comparable AP displacement cut-off stemming from the +30% and +50% analysis, the +50% ∆Area increase threshold shows both a satisfactory number of UA slices positively responding to ROC curves and represents a clinically substantial UA ∆Area increase. In our opinion, this renders it the most clinically relevant one, and subsequently the one we used to comprehensively summarise the results.

The findings above are compatible with the existence of the neutral-impact interval of which we theorized the existence, and its upper bound appears to be located around the +5 mm AP displacement mark. However, the sample included only mandibular setbacks lesser or equal to −5.8 mm and a comparison with greater negative mandibular displacement could not be carried out. Due to the lack of setbacks greater than 5 mm in our sample, we did not manage to determine the lower bound of the neutral-impact interval, of which we hypothesised the existence.

A limitation of our study is the analysis of the sole antero-posterior repositioning in relation to UA morphology, due to the complexity associated with the interpretation of multi-varied analyses. It is widely recognised that jaw rotational movements, especially counterclockwise rotation, can affect the volume of the UA, hence the possibility of an analysis of the sole AP movement to overestimate the effect induced by sagittal jaw repositioning. Further work on the matter could focus on the description of the effect of maxillo-mandibular rotations and vertical translation on UA morphology, together with the expansion of the patient population to confirm the present findings. Additionally, a future perspective for this method could be an implementation with computed fluid dynamics in silico experiments on the single patient model, as was done by Borojeni et al. for the nasal cavity [25].

## 6. Conclusions

The method proved to be reliable and reproducible. According to our findings MMA and MAMS do not affect the upper airways in a significantly different pattern when skeletal antero-posterior translation is inferior to 5 mm, when considering both cross-sectional area and total volume variations. These results suggest the existence of a neutral-impact interval with extension from −5 mm to +5 mm of antero-posterior displacement (with negative values only attributable to mandibular setback) in which the UA soft tissues modification appears to compensate for the skeletal displacement. Within this range the UA morphologic variation is unpredictable, with an overall tendency to maintain the preoperative condition.

All in all, a mandibular setback of 5 mm or less, with a concomitant varying degree of maxillary advancement, can be considered a safe procedure to perform in regard to the UA morphology. Additionally, for orthognathic surgery to induce +50% ∆Area increase of all three subregions of the pharynx, skeletal anterior–posterior translation requires overcoming a threshold point of +5.53 mm on average. A greater sample size is needed to further confirm the reported findings and complement them in terms of how maxillo-mandibular rotational and vertical displacements may affect UA morphology.

## Figures and Tables

**Figure 1 jpm-11-00177-f001:**
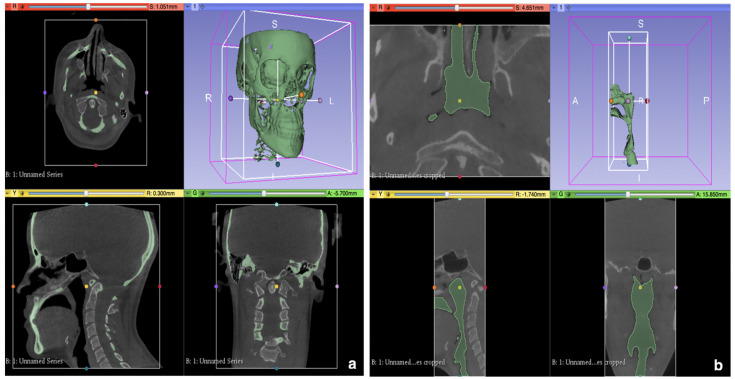
(**a**): Example of skull and maxillary bones segmentation in 3D Slicer software. (**b**): Example of upper airways segmentation in 3D Slicer software.

**Figure 2 jpm-11-00177-f002:**
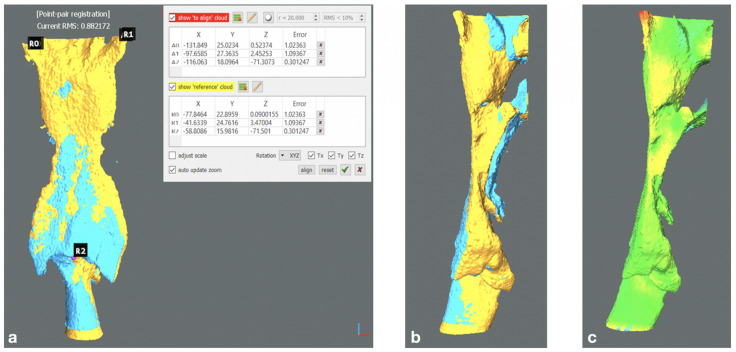
Superimposition of post- to pre-operative upper airways 3D models following point-pairs picking alignment in CloudCompare software. (**a**) Posterior view. (**b**) Sagittal view. (**c**) Colorimetric surface map following iterative closest point (ICP) to refine alignment in CloudCompare software (Sagittal view).

**Figure 3 jpm-11-00177-f003:**
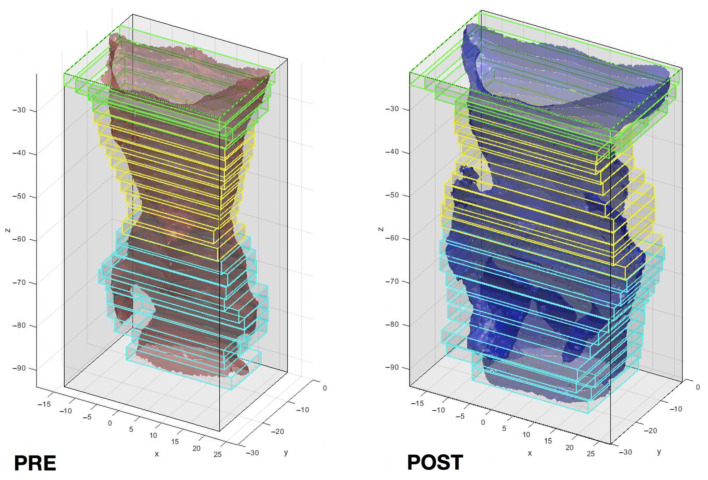
Upper airways 3D models evaluation and partitioning into bounding boxes in Matlab software. One single case is reported (**left**: pre-operative, **right**: post-operative).

**Figure 4 jpm-11-00177-f004:**
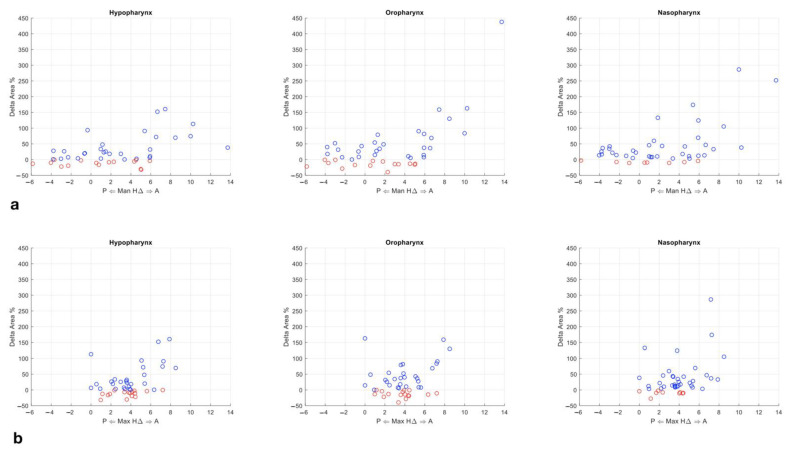
Plotted data of upper airways (UA) ∆Area variation in relation to maxillary bones displacement. (**a**) The X axis reports the maxillary antero–posterior repositioning (AP) movement, while the *Y*-axis reports the average percentage ∆Area variation of the slices of the corresponding patient for each subregion. Red circles indicate an average percentage reduction in area while blue circles indicate an average increase. (**b**) The X axis reports the mandibular AP movement, while the *Y*-axis reports the average percentage ∆Area variation of the slices of the corresponding patient for each subregion. Red circles indicate an average percentage reduction in area while blue circles indicate an average increase.

**Figure 5 jpm-11-00177-f005:**
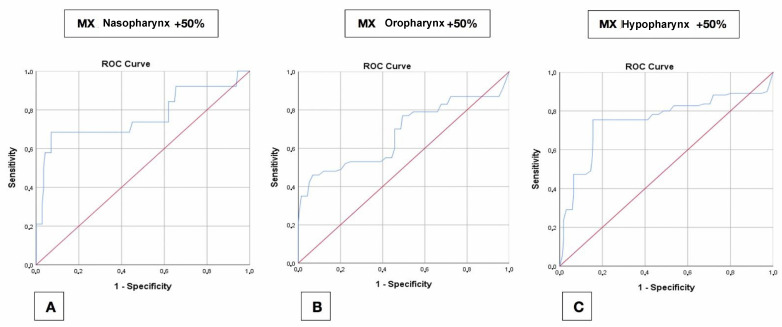
ROC curves relative to the +50% area increase; maxillary displacement for the three subregions analysed. (**A**) Nasopharynx. (**B**) Oropharynx. (**C**) Hypopharynx.

**Figure 6 jpm-11-00177-f006:**
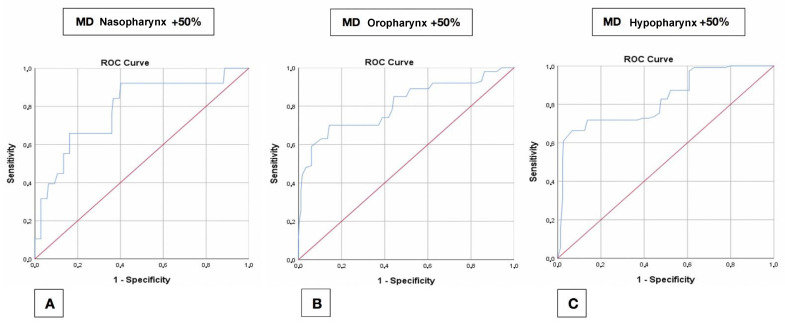
ROC curves relative to the +50% area increase; mandibular displacement for the three subregions analysed. (**A**) Nasopharynx. (**B**) Oropharynx. (**C**) Hypopharynx.

**Table 1 jpm-11-00177-t001:** Descriptive statistics of maxillary and mandibular displacement in each considered subgroup.

		N	Max	Min	Mean	S.D.	Median	IQR
**MAMS**	MX AP	16	7.20	0.90	3.58	1.66	3.90	2.40
	MD AP	16	0.00	−5.80	−2.39	1.63	−2.50	3.00
**MMA**	MX AP	29	15.10	0.00	4.36	3.07	3.80	3.50
	MD AP	29	13.70	0.50	4.66	3.25	4.60	4.55
**MMA—MX < 5**	MX AP	19	4.40	0.00	2.69	1.42	3.40	2.10
	MD AP	19	10.20	0.50	3.49	2.52	3.00	3.70
**MMA—MX ≥ 5**	MX AP	10	15.10	5.20	7.52	2.88	7.00	2.52
	MD AP	10	13.70	1.10	6.86	3.46	6.60	3.98
**MMA—MD < 5**	MX AP	16	6.30	0.50	3.39	1.38	3.50	1.65
	MD AP	16	5.00	0.50	2.38	1.54	1.85	3.05
**MMA—MD ≥ 5**	MX AP	13	15.10	0.00	5.54	4.1	5.60	5.30
	MD AP	13	13.70	5.00	7.46	2.51	6.50	3.35

S.D.: Standard Deviation; IQR: Interquartile Range.

**Table 2 jpm-11-00177-t002:** Descriptive statistics of ∆Area of UA slices in each considered subgroup, tabulated by subregion.

	Delta Area %												
	Hypopharynx				Oropharynx				Nasopharynx				*n* Patients
	Mean	SD	min	Max	Mean	SD	min	Max	Mean	SD	min	Max	
**MAMS**	10	35	−37	176	12	27	−36	91	18	22	−10	118	16
**MMA**	37	66	−49	317	54	91	−57	769	44	61	−70	336	29
**MMA—Mn < 5 mm**	6	30	−44	176	12	37	−57	221	23	48	−70	336	16
**MMA—Mn ≥ 5 mm**	74	79	−49	317	106	142	−22	769	70	71	−13	272	13
**MMA—Mx < 5 mm**	12	42	−49	176	24	50	−57	221	24	45	−66	336	19
**MMA—Mx ≥ 5 mm**	84	77	−26	317	110	159	−17	769	81	77	−70	272	10

**Table 3 jpm-11-00177-t003:** Maxillary and mandibular AP displacement (∆AP) threshold points identified by receiver operating characteristic (ROC) curves for each considered ∆Area increase and subregion. Youden’s indexes corresponding to each threshold point and ROC curve area under curve (ROC AUC) are also reported.

		Nasopharynx			Oropharynx			Hypopharynx		
	Δ Area	Δ AP (mm)	Youden’s Index	ROC AUC	Δ AP (mm)	Youden’s Index	ROC AUC	Δ AP (mm)	Youden’s Index	ROC AUC
Maxilla	30%	5.5	0.478	0.702	6.6	0.287	0.629	4.8	0.477	0.700
	50%	5.5	0.614	0.766	6.6	0.391	0.677	4.8	0.599	0.753
	100%	6.6	0.815	0.890	7.0	0.597	0.735	4.8	0.566	0.728
Mandible	30%	(1.4) 5.2	(0.407) 0.380	0.723	5.2	0.384	0.687	5.9	0.494	0.746
	50%	(1.4) 5.2	(0.520) 0.496	0.785	5.2	0.557	0.805	5.9	0.596	0.826
	100%	5.2	0.630	0.810	5.2	0.751	0.935	6.2	0.776	0.902

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
