# Peer review of "Existence of a Neutral-Impact Maxillo-Mandibular Displacement on Upper Airways Morphology"

_jpm, 2021, doi:10.3390/jpm11030177_

Round 1

Reviewer 1 Report

The idea to investigate whether MMA and MAMS elicit a significantly different impact on the UA morphology and to determine the existence and extension of a neutral-impact interval of maxillary and mandibular displacement, within which the UA morphology exhibits only minor and unpredictable variations is innovative and interesting. I also think that it is useful from a clinical point of view. The analysis of the displacement only in the AP direction has some limitations which are well explained in the discussion by the authors.

The article is well structured and written. I have few comments:

  • The t test cannot be used to evaluate percentages, so to evaluate the UA total volume differences you should use a non-parametric test.
  • In table 1, I think that the min and max displacements values are inverted.
  • The analysis of the displacement only in the AP direction has some limitations which are well explained in the discussion by the authors.

There are also some minor mistakes:

Page 2 line 59: circa should be translated in English

Page 10 lines 286-290: I would add Table 3 at the end of the sentence

I prefer to write cut-off in this way instead of cutoff as is used in the manuscript

Author Response

We thank the reviewer for their in-depth work and positive comments.

  • We have conducted the more appropriate nonparametric Mann-Whitney test as suggested by the reviewer and subsequently revised the manuscript to include the results. The new findings confirmed those previously obtained with the Student t-test: a statistical difference was found only for MAMS vs MMA with Upper Maxilla AP greater or equal to 5mm, and for MAMS vs MMA with mandible AP greater or equal to 5mm.
  • Table 1 was correctly formatted.
  • As stated in the manuscript, only the analysis of the AP displacement was reported, due to the complexity of multi-varied analyses and the sake of readability itself.
  • Minor mistakes were revised.

Reviewer 2 Report

This is a good study to investigate the existence and extension of a neutral-impact interval of bony segments displacement on the upper airways morphology.

The manuscript is useful to provide researchers information of existence of a neutral-impact maxillo-mandibular displacement on upper airways morphology. However, there are some questions need to be addressed before publication. Below are specific minor comments:

1. In Introduction section: Refer to the recent study, which presents method of image segmentation and reconstruction of upper airway geometry based on CBCT scans:

- Borojeni et al. (2020), Normative ranges of nasal airflow variables in healthy adults,
International Journal of Computer Assisted Radiology and Surgery, volume 15, pages 87-98 (2020)

2. Provide the subject information such as age, gender, weight, and height in the manuscript.

3. How did the authors validate their results? It is necessary to compare the results with the previously published data from the literature.

Author Response

We thank the reviewer for their in-depth work, the positive comments and interesting suggestions. 

  • The study by Borojeni et al brought to attention by the reviewer is very well conducted and shows, to a certain extent, similarities with our work. However, it focuses on the nasal cavity region and is conducted in silico. We thank the reviewer for the suggestion, as this could prove to be interesting for our practice. We chose to quote the study at the end of the discussion section.
  • Due to the premises of the informed consent signed by patients, data regarding patient-specific parameters such as age and weight can be published only as aggregated data (i.e. descriptive statistics). Additionally, weight and height information could not be retrieved as the vast majority (~90%) of the patients sample were not treated for OSA, therefore such parameters were not considered strictly necessary.
  • With the present study we authors aimed to introduce a novel analysis method for the evaluation of the upper airways morphology, following an extensive inspection of the literature. We in fact tested several methods published in the past, which, however, resulted in non-negligible methodological biases. For instance, we tested the workflow proposed by Tan et al and Lee et al, whom we cited in the discussion section and whose academic work we greatly admire, but found them to be inapplicable due to flaws related to anatomical landmarks picking, which would not allow to superimpose post- to pre-operative 3D models. On the basis that mentioned methods could not be employed due to criticalities, we created a different semi-automated 3D method, therefore a comparison of the results could not be carried out.  
